# Pre-Workout Intake of High-Protein Products: Palatability and Gastrointestinal Effects of Milk vs. Yogurt

**DOI:** 10.3390/nu17223540

**Published:** 2025-11-12

**Authors:** Priscilla Portuguez-Molina, Julián Camilo Garzón-Mosquera, Luis Fernando AragónVargas

**Affiliations:** 1Human Movement Science Research Center, University of Costa Rica, San José 11501-2060, Costa Rica; julian.garzon@ucr.ac.cr; 2School of Physical Education and Sports, Human Movement Science Research Center, University of Costa Rica, San José 11501-2060, Costa Rica; luis.aragon@ucr.ac.cr

**Keywords:** palatability, gastrointestinal discomfort, strength training, protein intake

## Abstract

**Background:** Muscle mass is a fundamental component for overall health and physical performance. A combination of exercise and protein intake is the best way to enhance protein synthesis and develop muscle mass. In order to promote protein intake, palatability and gastrointestinal symptoms should be considered when evaluating dairy products, such as milk or yogurt, as a viable, convenient and tolerable option before exercise. **Objective:** To compare the palatability of two protein-rich products before starting resistance exercise and potential gastrointestinal symptoms that may arise during exercise after their consumption. **Methods:** In a randomized, crossover study, 30 physically active individuals visited the laboratory on two occasions. During each visit, they consumed ≈25 g protein in the form of milk (500 mL) or yogurt (200 mL) and performed a resistance training session. The palatability of the products was assessed after consumption and perceived gastrointestinal symptoms were measured before product intake, after intake, at the middle and at the end of the exercise session. **Results:** Sweetness was higher for milk (7.73 ± 1.36) than for yogurt (7.13 ± 1.48) (*p* = 0.034). Overall acceptance was higher for milk as well (7.63 ± 1.09 for yogurt and 7.97 ± 0.85 for milk) (*p* = 0.048). Regarding GI symptoms, abdominal bloating and belching showed differences between products, being greater with milk, while thick saliva, reflux and nausea showed differences among measurement times. Overall, reported GI symptoms were few, and they were mostly mild for both products. **Conclusions:** Both products showed good acceptance and tolerance and therefore, can be considered good options for protein intake prior to resistance exercise.

## 1. Introduction

Muscle mass is a fundamental component for overall health and physical performance. Acquiring adequate muscle mass at earlier ages can improve an individual’s functional capacity and help to prevent deterioration and disability [1]. In the sports field, beyond performance-related benefits, protein consumption has been shown to reduce the catabolic response to exercise-induced muscle stress and to accelerate recovery [2,3].

Scientific literature has consistently shown that proper protein supplementation has a positive impact on athletic performance and, additionally, that exercise stimulates muscle protein synthesis. However, one of the most effective stimuli for protein synthesis is the combination of protein intake with exercise, as this produces a greater net anabolic effect [4].

Although the exact intake timing remains a subject of debate, current evidence indicates that total daily protein intake is a stronger predictor than the timing of consumption [5]. Therefore, recommendations suggest that total protein intake should be spaced out approximately every 3 h throughout the day in doses of 20–40 g of high-quality protein, emphasizing that the optimal timing depends on individual tolerance [4,6].

The effectiveness of protein-containing products depends on a variety of factors. The sensory acceptance of food directly influences voluntary consumption and, consequently, its long-term effectiveness. Hydration studies such as Burdon et al. [7] showed that greater palatability was associated with higher consumption and, as a result, improved hydration and performance in participants. This same phenomenon has been identified in other research, where flavored beverages promoted voluntary fluid intake, improved fluid balance, and prevented dehydration, when compared to water [8,9,10].

However, the limited number of studies specifically addressing the palatability of foods consumed close to exercise is noteworthy. This lack of attention may be related to the frequent occurrence of gastrointestinal (GI) issues resulting from the consumption of products prior to physical activity. Although GI symptoms are more common in endurance sports [11], authors such as Hart et al. [12] have shown that GI discomfort can also occur during resistance exercise. When the individual effect of exercise is combined with prior protein consumption, symptoms may be exacerbated.

After many studies have clarified the importance of protein consumption in active and general population, the gap between theoretical nutritional guidelines and real-world applications should be reduced. While sports nutrition often provides general recommendations on macronutrient timing and intake, there is limited evidence on whether commonly consumed products are both palatable and tolerable when used in daily life.

Therefore, the purpose of this study was to compare the palatability of two protein-rich products consumed before resistance exercise, as well as to evaluate the gastrointestinal symptoms that arose during exercise following their consumption. The aim was to develop effective and tolerable strategies to promote voluntary protein intake before resistance exercise.

## 2. Materials and Methods

A total of 30 apparently healthy, physically active subjects aged between 18 and 45 years volunteered to participate in this study. The Physical Activity Readiness Questionnaire (PAR-Q) [13] was used to ensure participants could safely engage in physical activity; and they had to meet the American College of Sports Medicine [14] recommendations of engaging in at least 150 min per week of moderate-intensity physical activity or at least 75 min per week of vigorous-intensity physical activity. Each participant read and signed the informed consent form before beginning the study. The study protocol was approved by the Scientific Ethics Committee of the University of Costa Rica, in accordance with official letter CEC-655-2023, on 30 October 2023, and in compliance with the Declaration of Helsinki.

This was a randomized, crossover study. Prior to laboratory arrival, participants were asked to fast for at least 3 h and to refrain from engaging in strenuous exercise on the testing day and the day before. Participants reported to the laboratory on two separate occasions, one for each product, with athletic clothing, and were measured for body weight and height. During both sessions, the same procedures were followed, and the only difference was the product intake. Participants consumed the assigned product (randomized order), and were asked: On a scale of 1 to 10, how full do you feel? Afterward, they completed a gastrointestinal symptoms questionnaire for the first time (Pre-consumption) and proceeded to consume a commercially available protein supplement: either Greek yogurt (flavors: coconut, almond, strawberry-banana, or blueberry; 200 mL, 25 g protein, 153 kcal, 8.7 g carbohydrates, 4 g fat, 0.3 g lactose [lactose-free]) or milk (flavors: vanilla, strawberry, or chocolate; 500 mL, 24 g protein, 212 kcal, 24 g carbohydrates, 4 g fat, <0.6 g lactose [lactose-free]). Cow’s milk was used in both milk and yogurt; and they were labeled as lactose-free. The protein content was matched between the two products, though the yogurt had a higher protein density (grams per mL), while the milk contained a higher total volume and carbohydrate content. Both products were served at 15 °C and consumed in their entirety. However, participants were instructed to consume the product at their preferred pace. As a result, some individuals ingested the full portion at once, while others consumed it gradually in small sips, according to their usual habits or tolerance. Consumption time was recorded and ranged from 39 to 1264 s. After consuming the assigned product (flavor of their choice), participants completed the palatability scale, and the gastrointestinal symptoms scale (post-consumption). The exercise session then began immediately after, and the GI symptoms scale was completed again at the middle (mid-session), during the 2 min breaks, and at the end of the session (End).

Palatability was assessed using a 9-point hedonic scale, where 1 corresponded to “extremely dislike” and 9 to “extremely like” [15]. A total of 17 gastrointestinal and systemic symptoms were recorded using a 10-point scale, where 0 indicated “no problem at all” and 9 indicated “the worst I’ve ever felt.” The symptoms on the scale were divided into three categories: upper gastrointestinal tract, lower gastrointestinal tract, and systemic symptoms [16]. For interpretability, and to align with prior work [17,18,19]. We categorized severity as follows: 0 = no symptom; 1–3 = mild; 4–6 = moderate; 7–9 = severe.

The exercise protocol consisted of a resistance training session based on the FITT recommendations of ACSM [14]. A warm-up was performed prior to the protocol, and stretching was conducted at the end. Each participant completed 3 sets of 10 repetitions for each exercise, at an intensity of 6–7 on the Borg Rating of Perceived Exertion scale (1–10) [20]. The exercise blocks were organized into the following supersets: (a) squat and chest press, (b) lunge and row, (c) leg curl and shoulder press, (d) leg extension and triceps extension. After completing each pair of exercises, participants had a 2 min rest period. The exercise session was completed in approximately 45 min, and participants exercised indoors in the sports facilities of the University of Costa Rica. The room was not air-conditioned, but a fan was used to provide ventilation and air circulation. Environmental conditions reflected those of a typical day in San José, Costa Rica, with a temperature range of 23–25 °C and 70–80% relative humidity.

All participants were measured on two different days (one for yogurt and one for milk, randomized), separated by approximately one week.

### Data Analysis

Data was analyzed using the statistical software SPSS v.24. Tests for normality and homoscedasticity were conducted for each variable. Descriptive statistics were obtained for palatability and gastrointestinal symptom variables.

T-student analysis was performed for all gastrointestinal symptoms prior to the exercise session, as well as for perceived fullness, to verify that participants arrived under similar conditions for both sessions.

To compare the palatability of the two products, a one-way ANOVA was conducted. To compare gastrointestinal symptoms between both conditions, a repeated-measures ANOVA (condition × time point) was performed to determine whether there was an interaction between the condition (yogurt or milk) and the time point: Pre-consumption (Initial measurement, before consuming the product, Pre), post consumption (immediately after the product was consumed, Post), mid-session (measurement taken halfway through the exercise session) or end of the session (final measurement, after the exercise session was finished); or an individual effect of either condition or time point. If the interaction was significant, the different means at each time point were compared separately for yogurt and milk using a repeated-measures ANOVA. Additionally, at each time point, means of yogurt and milk were compared using a paired *t*-test.

If only main effects were identified, a one-way repeated-measures ANOVA was performed to assess the effect of time. A *p*-value < 0.05 was considered statistically significant.

## 3. Results

A total of 30 physically active participants took part in the study (20 men and 10 women), age was 25.6 ± 6.4 (mean ± SD) years, height = 164.3 ± 8.5 cm, weight = 65.45 ± 12.62 kg; most had prior experience with resistance training.

### 3.1. Palatability

Milk scored higher (7.7 ± 1.4) than yogurt (7.1 ± 1.5) for sweetness (*p* = 0.034), as well as for overall acceptance (7.6 ± 1.1 for yogurt and 8.0 ± 0.9 for milk) (*p* = 0.048). Ratings for both products were high for both variables (sweetness and overall acceptance). The variables saltiness, sourness, mouthfeel, and aroma did not show significant differences between products. Average values for each variable and product are presented in Table 1.

### 3.2. Gastrointestinal Symptoms

Only belching and bloating showed differences between products. Figure 1 illustrates the average values for each time point and product.

Thick saliva, reflux and nausea showed a significant difference across measurement time points. Figure 2, Figure 3 and Figure 4 illustrate the average values at each time point.

Heartburn, upper intestinal cramps, lower intestinal cramps, vomiting, right or left abdominal pain, flatulence, loose stools or diarrhea, urgency to defecate, cramps throughout the digestive system, headache, urge to urinate, and dizziness showed no significant differences between products or across the different measurement time points.

The symptoms reported by the highest number of participants were thick saliva (39 reports for yogurt and 37 for milk), belching (39 for yogurt and 43 for milk), and bloating (26 for yogurt and 34 for milk). This was based on symptoms reported at any of the three times after product consumption (post-consumption, mid-session or end). The remaining symptoms were reported fewer than 27 times.

Table 2 presents a classification of gastrointestinal symptoms as no symptom (0) mild (1–3), moderate (4–6), or severe (7–9). It can be observed that many of the participants reported no symptoms. Symptoms reported were mostly categorized as mild, followed by moderate, with a lower incidence of severe symptoms. Symptoms with the highest incidence in the severe category were thick saliva (3.3% for both products), belching (2.3% for both products) and bloating (6.7%) for milk).

## 4. Discussion

In line with the aim of the study, the main findings revealed that both milk and yogurt were well accepted and generally well tolerated when consumed prior to resistance exercise. Specifically, milk was rated significantly higher in sweetness and overall acceptance compared to yogurt. Although some gastrointestinal (GI) symptoms such as abdominal bloating and belching were more pronounced after milk intake, other symptoms like thick saliva, reflux, and nausea varied over time but were not product specific. Importantly, the overall incidence and severity of GI symptoms were low and mostly mild for both dairy products.

### 4.1. Palatability

The results of the present study show an appropriate acceptance of both products by the participants. Notably, milk scored higher in sweetness and overall acceptability, while being perceived similarly to yogurt in the other variables (saltiness, sourness, mouthfeel and aroma).

Sweetness is considered an important factor in the palatability of a food, and there is evidence consistent with the hypothesis that sweet taste can stimulate hunger and appetite, leading to greater voluntary consumption [21]. Sweetened beverages tend to be rated as more palatable than plain water, as demonstrated by King et al. [22], who found that participants preferred an artificially sweetened beverage after exercise rather than water. Similarly, Beridot-Therond et al. [23], reported higher consumption of two sweetened beverages compared to mineral water and a drink with only orange flavor, suggesting that sweetness alone can promote voluntary intake. This can be considered an important factor when it comes to the palatability of a food, and it must be considered when the protein intake of a participant throughout the day is studied.

In terms of overall acceptance, the relationship between the palatability of foods and their frequency of consumption is well stablished [24]: people tend to choose their intake options based on taste [25]. Flavored beverages are generally more palatable, which leads to greater voluntary consumption, better fluid balance, and therefore can help prevent dehydration [9]. A product with an optimal level of sweetness and an adequate palatability is likely to be consumed in greater quantities due to greater voluntary consumption, as stated by Rivera-Brown et al. [10] when they compared a sweetened and flavored drink to water during an ad libitum session after aerobic exercise.

Since the aim of this study is to identify viable options for physically active individuals to consume protein throughout the day, it is essential to promote intake through products they enjoy. Although the differences in sweetness and overall acceptance were statistically significant, the absolute values show that both products had a high level of liking (greater than 7 on a 1-to-9 scale). This suggests that both protein-rich yogurt and milk are viable options for pre-exercise consumption and offer two different alternatives based on individual preferences, tolerance, or availability.

### 4.2. Gastrointestinal Symptoms

Few studies have investigated the report of gastrointestinal symptoms during resistance exercise. The study by Hart et al. [12] is one of the few in this area, where the most prevalent symptoms reported were nausea and vomiting. Although the exercise protocol was similar in terms of sets, repetitions, and rest, Hart et al. [12] did not include any protein ingestion at all. The most common symptoms in the present study (thick or reduced saliva, belching, and bloating) may be associated with the specific properties of dairy products. For instance, when milk mixes with saliva, it increases viscosity and volume, which can affect sensory perception in some individuals [26].

When compared with the reports on cardiovascular exercise, the symptoms differ. A study conducted by Parnell et al. [27] addressed the most common GI symptoms experienced by runners through a questionnaire and the results included: abdominal pain/cramps, intestinal discomfort, urge to defecate, and bloating. Bloating was the common factor between both studies, while the other symptoms changed. It is important to highlight that, unlike endurance exercise, resistance training does not involve mechanical trauma or constant body movement and tends to be shorter in duration [28] and therefore associated symptoms can change between these two activities.

As it pertains to the reported symptoms in the present study, bloating and belching were the GI symptoms that differed between products, with milk being rated higher in both. In this case, participants were required to consume a larger volume of milk (500 mL, two boxes) to better match the protein content of the yogurt (200 mL), which may have influenced the observed differences. It is also important to note that both products are dairy-based and high in protein. Current recommendations often advise against protein intake before exercise due to its potential to induce gastrointestinal symptoms [28] and, on the other hand, in the study by Parnell et al. [27], many runners reported avoiding dairy during exercise because it caused them abdominal discomfort. However, in the present study, both products were well tolerated by the participants, suggesting they can be good options for pre-exercise protein consumption.

From a practical standpoint, the relationship between appetite and physical activity has been frequently studied, with several studies showing that moderate or vigorous exercise can reduce appetite [29], which reinforces the importance of consuming protein before exercise when the goal is to improve strength and body composition, since pre-exercise consumption reduces post-exercise requirements [6]. This is supported by the International Society of Sports Nutrition, stating that the optimal timing for protein intake depends on the individual’s tolerance [4], given that the beneficial effects have been demonstrated both before and after exercise.

As thick saliva, reflux and nausea increased consistently from baseline to the end of the measurements, but just slightly, we suggest a longer interval between product consumption and exercise onset, for future studies.

Lastly, it is important to highlight that various studies, such as Karhu et al. [30] have classified participants in gastrointestinal-related research as symptomatic or asymptomatic based on their personal history with such symptoms during exercise. This suggests that some individuals may be more prone to developing these symptoms. In symptomatic individuals, findings point to an improvement in symptom prevalence and severity with “training”, that is, with practice. Jeukendrup [31] explains that the gastrointestinal tract is adaptable, and various factors such as fullness perception, GI discomfort, and gastric emptying can change even after just a few sessions. Future studies should address this adaptation of the GI tract to regular intake of protein before resistance exercise, as well as focusing on exploring strategies that allow for an optimal, tolerable, and practical protein intake approach for physically active individuals.

## 5. Conclusions

Milk received higher scores than yogurt in terms of sweetness and overall acceptance; however, regarding gastrointestinal issues, it also showed higher values for belching and abdominal bloating. Regardless, both products were well accepted and tolerated by the participants, indicating that they can be used as part of a strategy for spaced protein intake throughout the day in physically active individuals, particularly before resistance exercise.

## 6. Practical Applications

Both protein-enriched milk and yogurt may be good options to increase protein intake throughout the dayConsuming protein before resistance training can be an option for athletes with multiple training sessions in a day or those who perform consecutive training sessions.Future studies should focus on long-term consumption to see if variables like palatability and gastrointestinal symptoms differ with repeated intake.The adaptation of the GI tract to regular protein intake before resistance exercise could be analyzed in future studies.

## 7. Limitations

In order to match protein, products had different consumption volumes—milk: 500 mL vs. yogurt: 200 mL—which affects some comparisons between them.Because of the study design there was no control group.A few participants had little experience with resistance exercise, which may have affected their perceived exertion and therefore their weight choices.Data was obtained through a self-reported questionnaire using a subjective scale, which may introduce perceptual bias and variability among participants. Future research may attempt to include additional, objective, variables.We did not formally test the participants for milk protein intolerance prior to the study, which can affect the symptoms reports. Future studies should report this information in order to ensure more accurate interpretation of gastrointestinal symptoms.Lower gastrointestinal symptoms might not have been comprehensively recorded given the brief observation window employed in this study (only approximately 1 h after product consumption). Since the primary aim of our symptom assessment was to capture immediate and short-term gastrointestinal responses during and shortly after product consumption in the context of pre-exercise intake, this was beyond the scope of the study.

## Figures and Tables

**Figure 1 nutrients-17-03540-f001:**
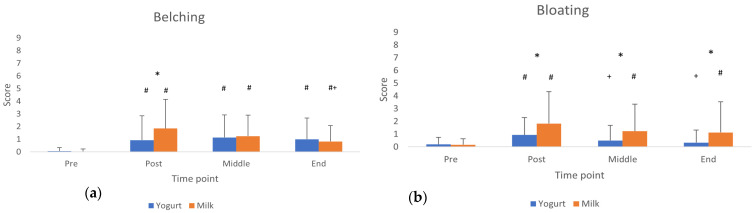
Belching (**a**) and bloating (**b**) values across measurement time points. * Difference between products (*p* < 0.05). # Difference from baseline (pre consumption). + Difference from Post Consumption (*p* < 0.05).

**Figure 2 nutrients-17-03540-f002:**
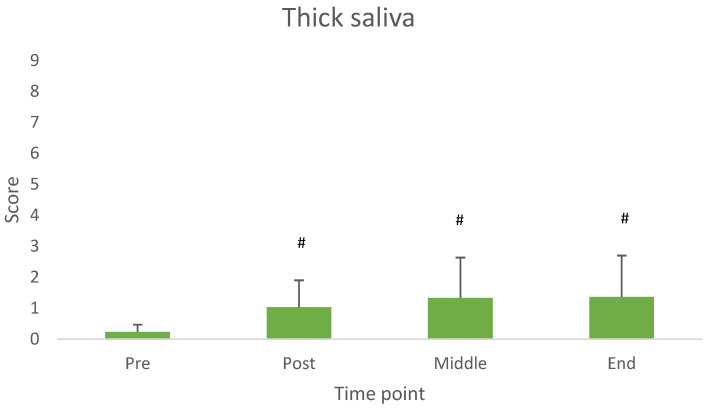
Values of thick saliva across measurement time points. Different from Pre-consumption (*p* < 0.05). # Difference from baseline (pre consumption) (*p* < 0.05).

**Figure 3 nutrients-17-03540-f003:**
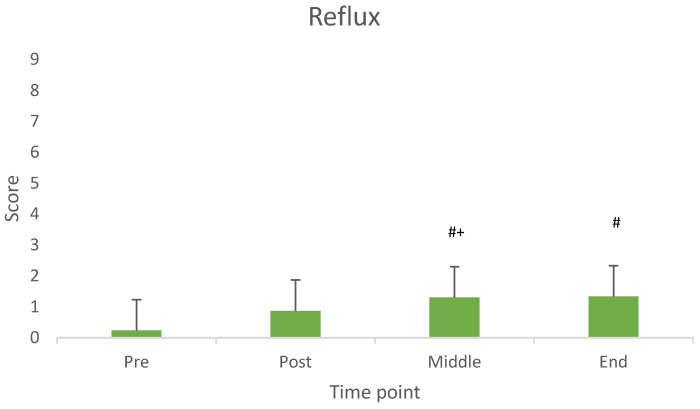
Values of reflux across measurement time points. # Difference from baseline (pre consumption) (*p* < 0.05). + Difference from Post Consumption (*p* < 0.05).

**Figure 4 nutrients-17-03540-f004:**
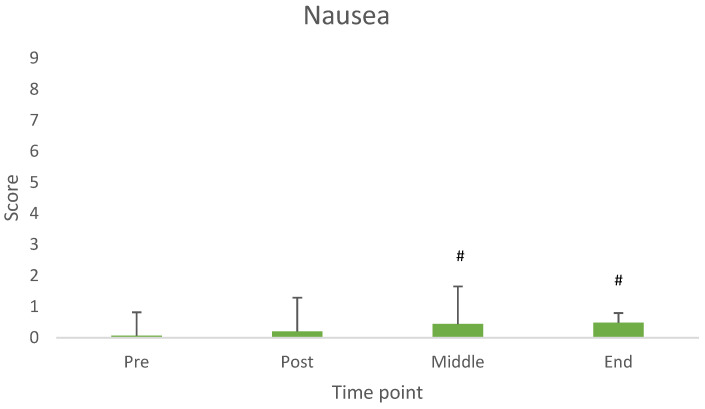
Values of nausea across measurement time points. # Difference from baseline (pre consumption) (*p* < 0.05).

**Table 1 nutrients-17-03540-t001:** Palatability scores for each product.

	Yogurt	Milk	*n*	*p*
	$(\bar{x}$ ± DS)	$(\bar{x}$ ± DS)		
Sweetness	7.1 ± 1.5	7.7 ± 1.4	30	0.034 *
Saltiness	6.0 ± 1.4	5.7 ± 1.9	29	0.326
Sourness	5.9 ± 1.6	5.7 ± 2.0	30	0.451
Mouthfeel	6.9 ± 1.5	7.2 ± 1.9	30	0.409
Aroma	7.3 ± 1.4	7.4 ± 1.6	30	0.887
Overall acceptance	7.6 ± 1.1	8.0 ± 0.9	30	0.048 *

Note: * Difference between products (*p* < 0.05).

**Table 2 nutrients-17-03540-t002:** Incidence (%) ^1^ classified by product and severity, for each gastrointestinal symptom.

Symptom	Yogurt				Milk			
	No Symptom	Mild	Moderate	Severe	No Symptom	Mild	Moderate	Severe
Reflux	72.2	26.7	1.1	0.0	70.0	23.3	5.6	1.1
Heartburn	86.7	13.3	0.0	0.0	90.0	8.9	1.1	0.0
Cramping	93.3	6.7	0.0	0.0	87.8	10.0	2.2	0.0
Vomiting	96.7	3.3	0.0	0.0	91.1	5.6	2.2	1.1
Nausea	86.5	13.5	0.0	0.0	80.0	16.7	2.2	1.1
Thick saliva	56.7	31.1	8.9	3.3	58.9	30.0	7.8	3.3
Belching	57.3	33.7	6.7	2.3	51.7	36.0	10.1	2.3
Intestinal cramps	86.7	12.2	1.1	0.0	90.0	8.9	1.1	0.0
Bloating	71.1	24.4	4.4	0.0	62.2	20.0	11.1	6.7
Abdominal pain	96.7	3.3	0.0	0.0	96.6	2.3	1.1	0.0
Flatulence	90.0	10.0	0.0	0.0	84.4	15.6	0.0	0.0
Loose stools/diarrhea	100.0	0.0	0.0	0.0	95.6	4.4	0.0	0.0
Urge to defecate	95.5	4.5	0.0	0.0	94.4	5.6	0.0	0.0
Muscle cramping	93.3	6.7	0.0	0.0	93.3	6.7	0.0	0.0
Headache	93.3	3.3	1.1	2.2	86.7	13.3	0.0	0.0
Urge to urinate	84.4	14.4	1.1	0.0	83.3	12.2	4.4	0.0
Dizziness	87.8	11.1	1.1	0.0	89.9	11.1	0.0	0.0

^1^ Percentage of times the symptoms appeared in relation to the 90 measurements (3 for each participant).

## Data Availability

The original data presented in the study is openly available in Kerwá at https://hdl.handle.net/10669/102917 (accessed on 30 October 2025).

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
