# Peer review of "Pre-Workout Intake of High-Protein Products: Palatability and Gastrointestinal Effects of Milk vs. Yogurt"

_nutrients, 2025, doi:10.3390/nu17223540_

Round 1

Reviewer 1 Report

Comments and Suggestions for Authors

The study presented has a very low degree of reliability and reproducibility. It is a comparison made in a single test, with only two foods. A moderate number of people participated (30 people in total), and all the results were obtained by completing a questionnaire, without measuring any clinical, biochemical, or anthropometric variables. The questions were asked using a subjective scale of 1 to 9 so that respondents could estimate the degree of compliance with the questions asked.
Without underestimating the value of studies based on surveys and questionnaires, it is true that these have a much higher degree of variability than results that can be assessed objectively.

Therefore, with a sample size of 30 participants, and given that the data obtained is based on a questionnaire whose perception by the subjects is always subjective, the results obtained are not very reliable.
The results are presented in a way that could be greatly improved. I do not consider that the work carried out contributes a sufficient degree of novelty to warrant publication of its results in a Q1 journal.

Author Response

Comment 1:

The study presented has a very low degree of reliability and reproducibility. It is a comparison made in a single test, with only two foods. A moderate number of people participated (30 people in total), and all the results were obtained by completing a questionnaire, without measuring any clinical, biochemical, or anthropometric variables. The questions were asked using a subjective scale of 1 to 9 so that respondents could estimate the degree of compliance with the questions asked.

Without underestimating the value of studies based on surveys and questionnaires, it is true that these have a much higher degree of variability than results that can be assessed objectively.

Therefore, with a sample size of 30 participants, and given that the data obtained is based on a questionnaire whose perception by the subjects is always subjective, the results obtained are not very reliable.

The results are presented in a way that could be greatly improved. I do not consider that the work carried out contributes a sufficient degree of novelty to warrant publication of its results in a Q1 journal.

Response 1:

Dear Reviewer,

We appreciate your thoughtful assessment. Our study was deliberately designed as a randomized, two-visit, within-subject crossover focused on two applied outcomes—palatability (9-point hedonic scale) and perceived gastrointestinal (GI) symptoms (0–9 scale)—which, by definition, require self-report and are widely used in exercise nutrition and sensory research to inform practice.

Objectivity: We acknowledge the limitations of this study and other studies based only on subjective questionnaires, particularly in terms of reliability. However, our main goal was not to assess clinical, biochemical or anthropometric variables, but rather to evaluate subjective perceptions of palatability and gastrointestinal comfort, which are assessed through self-reported scales. We would like to emphasize that the primary aim of our study is to bridge the gap between theoretical nutritional guidelines and real-world application. While sports nutrition often provides general recommendations on macronutrient timing and intake, there is limited evidence on whether commonly consumed products are both palatable and tolerable when used in daily life. Our study seeks to provide practical, evidence-based insights for athletes and active individuals by evaluating whether commonly used high-protein dairy options—such as milk or yogurt—are truly feasible as pre-workout foods, not only in terms of nutritional content, but also regarding gastrointestinal comfort and acceptability during physical effort. We made the corresponding changes in lines 65-69 to emphasize the importance of this article as we believe this applied perspective is essential to help nutrition professionals move from theoretical planning to personalized, actionable strategies that fit athletes’ preferences, tolerances, and performance needs.

Measurement tools: The 9-point hedonic scale and a 0–9 GI symptom scale are standard instruments in this literature and have demonstrated utility for detecting meaningful differences that translate into nutritional recommendations (Gabriel et al., 2005; Pugh et al. 2018; Tielemans et al., 2013; van Kerkhoven et al., 2008). We now provide anchors and severity cut-offs (0 = none; 1–3 = mild; 4–6 = moderate; 7–9 = severe) in the Methods Section (lines 111-116) to enhance interpretability

Sample size and design choice: While we agree that the sample of 30 participants is moderate, this number is consistent with similar crossover studies in the field of sports nutrition and sensory science, particularly when dealing with within-subject comparisons. Crossover design increases statistical power by reducing inter-subject variability, as each participant serves as their own control.

Novelty and contribution: We revised the Introduction to better position the gap: there is limited evidence on whether commonly consumed, lactose-free high-protein dairy matrices (commercial milk vs Greek yogurt) are both palatable and tolerable when ingested immediately before resistance exercise. Our findings provide applied, hypothesis-generating evidence to guide spacing of protein intake using real-world products.

In sum, we have strengthened transparency (design clarity, sample size, categorization), reframed the contribution as applied and hypothesis-generating, and outlined concrete methodological enhancements for future work (Methods and Limitation Sections).

References:

Gabriel, A. A., Fernandez, C. P., & Tiangson-Bayaga, C. L. P. (2005). Consumer acceptance of Philippine orange drink as an iron-fortified beverage for Filipino women. Food Science and Technology Research11(3), 269-277.

Pugh, J. N., Kirk, B., Fearn, R., Morton, J. P., & Close, G. L. (2018). Prevalence, severity and potential nutritional causes of gastrointestinal symptoms during a marathon in recreational runners. Nutrients10(7), 811.

Tielemans, M. M., Jaspers Focks, J., van Rossum, L. G., Eikendal, T., Jansen, J. B., Laheij, R. J., & van Oijen, M. G. (2013). Gastrointestinal symptoms are still prevalent and negatively impact health-related quality of life: a large cross-sectional population-based study in The Netherlands. PloS one8(7), e69876.

van Kerkhoven, L. A., Eikendal, T., Laheij, R. J., van Oijen, M. G., & Jansen, J. B. M. J. (2008). Gastrointestinal symptoms are still common in a general Western population.

Reviewer 2 Report

Comments and Suggestions for Authors

Pre-workout intake of high-protein products: Palatability and  Gastrointestinal Effects of Milk vs Yogurt

Experimental randomized, cross-sectional study to compare the palatability of two protein-rich products before starting resistance exercise and potential gastrointestinal symptoms that may arise during exercise after their consumption. 30 physically active individuals visited the laboratory on two occasions. During each visit, they consumed milk (500ml) or yogurt (200ml) (randomized) and performed a resistance training session. The limited number of studies specifically addressing the palatability of  foods consumed close to exercise is noteworthy. This study may help to address more properly this question.

My major question should be why the authors did not perform a cross-sectional study repeating the measurements with the other product in  all subjects, as there may be a bias of acceptability when comparing between different products.

Other major  comments:

  1. As volume differs between both groups this may explain partially the difference in GI symptoms (Gastric emptying generally increases with higher initial meal volume but then slows over time, a mechanism influenced by the gastric wall's stress and active tone.) We recommend to compare using similar products both in volume and nutrient load.
  2. Figures 2-4 are related to the whole group? Please, specify by product.
  3. Specify how the intensity of symptoms was evaluated. Cut-off points

Author Response

Comment 1:

My major question should be why the authors did not perform a cross-sectional study repeating the measurements with the other product in all subjects, as there may be a bias of acceptability when comparing between different products.

Response 1:

Thank you for this important point. We apologize for the confusion caused by our wording. The study was not a simple cross-sectional comparison; it was a randomized, within-subject, two-visit crossover design. Each participant consumed both products on separate laboratory visits in randomized order, thereby allowing each person to serve as his/her own control and minimizing between-subject acceptability bias. We have revised the Abstract (line 19) and Methods (line 86) to explicitly state “randomized, crossover study” and clarified the visit structure and randomization procedure to avoid any ambiguity.

Comment 2:

As volume differs between both groups this may explain partially the difference in GI symptoms (Gastric emptying generally increases with higher initial meal volume but then slows over time, a mechanism influenced by the gastric wall's stress and active tone.) We recommend to compare using similar products both in volume and nutrient load.

Response 2:

We appreciate this recommendation, but since our main goal was to evaluate whether commonly used high-protein dairy options—such as milk or yogurt—are truly feasible as pre-workout foods, not only in terms of nutritional content, but also regarding gastrointestinal comfort and acceptability during physical effort; we decided to match the protein content in both products. In this way, people can choose commercially available products that contain approximately 25g of protein even if they have different volumes.  However, we acknowledge that this is a limitation of our study and prevents us from making specific comparisons regarding the volume and content of other macronutrients. Therefore, we have included it in the Limitations section at the end of the study, lines 299-300.

Comment 3:

Figures 2-4 are related to the whole group? Please, specify by product

Response 3:

Since we did not find differences between products, Figures 2-4 describe the behavior of the whole group, which is the same as the behavior for each one of the products.

Comment 4:

Specify how the intensity of symptoms was evaluated. Cut-off points

Response 4:

Thank you for pointing this out. Briefly, gastrointestinal symptoms were assessed using a 10-point Likert-type scale with two explicit anchors: 0 = “no problem at all” and 9 = “the worst I’ve ever felt.” For interpretability and to align with prior work, we categorized severity as follows:

  • 0 = no symptom
  • 1–3 = mild
  • 4–6 = moderate
  • 7–9 = severe

These categories are now reported consistently in the Methods Section. We also cite prior usage of similar 0–9/0–10 GI symptom scales in exercise/nutrition settings to support this approach (lines 111-116)

Reviewer 3 Report

Comments and Suggestions for Authors

The authors compared the palatability of two commercially available high-protein products (yoghurt and milk) and assessed gastrointestinal symptoms occurring during resistance training after consumption of these products. The study was conducted in accordance with accepted principles and with the approval of the Ethics Committee.
The biggest drawback of this study is the lack of a control group, but it does provide important information about the reaction to milk components and promotes milk consumption as a source of protein.
I have a few comments on the manuscript:
Were the study participants tested for milk protein intolerance prior to the experiment? The symptoms of milk intolerance are varied and can affect the digestive system (abdominal pain, diarrhoea, vomiting, flatulence), the skin (rash, hives, itching), as well as the respiratory system (cough, wheezing, runny nose) and the nervous system (mood deterioration, sleep disorders). Therefore, this is important information.
What kind of milk was used in the studies: cow's, sheep's, goat's...?
What kind of milk was used to make the Greek yoghurt?
What was the temperature of the milk/yoghurt at the time of consumption? Was the milk/yoghurt divided into smaller portions or was it consumed in its entirety?
Lines 81-82. Study participants were asked to fast for at least 3 hours. Why such a short time? It is recommended that such studies be conducted after ten hours of fasting or longer.
Lines 100-101. The authors wrote: ‘The symptoms on the scale were divided into three categories: upper gastrointestinal tract, lower gastrointestinal tract, and systemic symptoms.’ Milk takes at least 5 hours to digest before its components reach the lower gastrointestinal tract. The authors do not mention anything about such a study after 5 hours. The authors should explain this, as the graphs show different time points: ‘Pre, Post, Middle, End’. The authors need to standardise the description, as it is misleading for the reader.
Lines 103-111. How long did the exercise session last (minutes, hours)? Under what conditions did the exercise session take place (temperature, air conditioning, indoors, outdoors, time of day, etc.)?
What does the term ‘mid-session’ mean in relation to the examination of gastrointestinal symptoms? Was the exercise interrupted to complete the questionnaire? Did the participants complete it during the 2-minute breaks?
Lines 179-181. The aim of the study is already stated in the introduction. The authors should remove this text.
Chapters 6 and 7 should be combined with the conclusions, adjusting the content accordingly.

Author Response

Comment 1:

The biggest drawback of this study is the lack of a control group, but it does provide important information about the reaction to milk components and promotes milk consumption as a source of protein.

Response 1:

Thank you for this important point. The study was not a simple cross-sectional comparison; it was a randomized, within-subject, two-visit crossover design, in which each participant acted as their own control and minimizing between-subject acceptability bias. Since the aim of the study was to provide practical, evidence-based insights for athletes and active individuals by evaluating whether commonly used high-protein dairy options are truly feasible as pre-workout foods, this design allowed us to directly compare individual responses to two dairy products (milk and yogurt) under similar conditions. We have revised the Abstract (line 19) and Methods (line 86) to explicitly state “randomized, crossover study” to clarify the design.

Comment 2:

Were the study participants tested for milk protein intolerance prior to the experiment? The symptoms of milk intolerance are varied and can affect the digestive system (abdominal pain, diarrhoea, vomiting, flatulence), the skin (rash, hives, itching), as well as the respiratory system (cough, wheezing, runny nose) and the nervous system (mood deterioration, sleep disorders). Therefore, this is important information.

Response 2:

Thank you for raising this important point. Participants were not formally tested for milk protein intolerance prior to the study. However, the dairy products used (both milk and yogurt) were lactose-free, which minimizes the likelihood that gastrointestinal symptoms observed were related to lactose intolerance.

Additionally, none of the participants reported known milk allergies or habitual avoidance of dairy products. While this does not rule out undiagnosed milk protein intolerance, the overall mild nature and low frequency of gastrointestinal symptoms suggest that such intolerance, if present, was not a major factor in the study outcomes.

However, given the importance of this point, we have acknowledged this as a limitation and clarified it in the manuscript, in the Limitations Section (lines 307-309).

Comment 3:

What kind of milk was used in the studies: cow's, sheep's, goat's...? What kind of milk was used to make the Greek yoghurt?

Response 3:

Thank you for your comment. Cow's milk was used in both products: milk and yogurt. We added this observation to the manuscript and improved the product description in the Methods Section (lines 98-100)

Comment 4:

What was the temperature of the milk/yoghurt at the time of consumption? Was the milk/yoghurt divided into smaller portions or was it consumed in its entirety?

Response 4:

Both products were served at 15 °C and they were consumed in their entirety. However, people were asked to consume the product at the preferred pace. As a result, some consumed it all at once, while others did it in small sips, according to their usual habits or tolerance. Since this is important information regarding the methodology, we added a more descriptive paragraph with these details in the Methods section (lines 101-105).

Comment 5:

Lines 81-82. Study participants were asked to fast for at least 3 hours. Why such a short time? It is recommended that such studies be conducted after ten hours of fasting or longer.

Response 5:

Thank you for your valuable comment regarding the fasting duration prior to product consumption. We intentionally selected a minimum fasting period of 3 hours to simulate more realistic, everyday conditions in which individuals might consume dairy products before exercise. Our goal was to assess the palatability and gastrointestinal symptoms under conditions closer to habitual intake, rather than under strict prolonged fasting which may not reflect typical behavior. We selected 3 hours to avoid people arriving to the laboratory feeling “full.

Additionally, upon arrival, participants rated their fullness on a scale from 1 to 10, and we analyzed whether their fullness influenced the outcomes. No significant differences were found between fullness levels and the measured variables, indicating that varying pre-consumption fullness did not affect the study results. This information is now available in the manuscript lines 92, 135-137)

Comment 6:

Lines 100-101. The authors wrote: ‘The symptoms on the scale were divided into three categories: upper gastrointestinal tract, lower gastrointestinal tract, and systemic symptoms.’ Milk takes at least 5 hours to digest before its components reach the lower gastrointestinal tract. The authors do not mention anything about such a study after 5 hours. The authors should explain this, as the graphs show different time points: ‘Pre, Post, Middle, End’. The authors need to standardize the description, as it is misleading for the reader.

Response 6:

Thank you for this important observation regarding the timing of milk digestion and symptom assessment. We acknowledge that milk components generally take several hours, around 5 hours—to reach the lower gastrointestinal tract. However, the primary aim of our symptom assessment was to capture immediate and short-term gastrointestinal responses during and shortly after product consumption in the context of pre-exercise intake.

The time points labeled as “Pre,” “Post,” “Middle,” and “End” correspond to symptom evaluations conducted before product consumption, immediately after, in the middle of the exercise session, and at the end of the exercise session, respectively. We recognize that these time points do not encompass the longer digestion period required for milk to reach the lower gastrointestinal tract, and thus our assessment primarily if pre-workout protein consumption was a tolerable option before exercise and not long-term outcomes.

We agree that the description could be clearer, and we have revised the manuscript to standardize and clarify the timing. We also added the description of each time point on lines 141-145 in Data Analysis: Pre-consumption (Initial measurement, before consuming the product, Pre), post consumption (immediately after the product was consumed, Post), mid-session (measurement taken halfway through the exercise session) or end of the session (final measurement, after the exercise session was finished. Finally, we have explicitly noted the limitation that lower gastrointestinal symptoms may not be fully captured within the short observation window used in this study (lines 310-314).

Comment 7:

Lines 103-111. How long did the exercise session last (minutes, hours)? Under what conditions did the exercise session take place (temperature, air conditioning, indoors, outdoors, time of day, etc.)?

Response 7:

We added the following paragraph to the Methods Section (lines 124-128), which describes in more detail the exercise session:

The exercise session was completed in approximately 45 minutes, and participants exercised indoors in the sports facilities of the University of Costa Rica. The room was not air-conditioned, but a fan was used to provide ventilation and air circulation. Environmental conditions reflected those of a typical day in San José, Costa Rica, with a temperature range of 23–25°C and 70–80% relative humidity.

 Comment 8:

What does the term ‘mid-session’ mean in relation to the examination of gastrointestinal symptoms? Was the exercise interrupted to complete the questionnaire? Did the participants complete it during the 2-minute breaks?

Response 8:

We wanted to find out whether people are affected by consuming milk or yogurt during exercise and also, to document if any symptoms appeared midway through the session, but with short duration and mild intensity that still allowed them to complete the session.

Participants completed the questionnaire during the 2 min breaks, while resting. This explanation was added to the manuscript in line 109.

Comment 9:

Lines 179-181. The aim of the study is already stated in the introduction. The authors should remove this text.

Response 9:

Text was changed to avoid repeating the aim of the study, but introducing the discussion with the main findings, to help guide the reader through the interpretation and relevance of the results.

Comment 10:

Chapters 6 and 7 should be combined with the conclusions, adjusting the content accordingly.

Response 10:

Thank you for the valuable suggestion to combine Chapters 6 and 7 (Practical Applications and Limitations) with the Conclusions. While we understand the rationale behind this recommendation, we believe that maintaining a separate section dedicated to Practical Applications is important because it highlights the direct, real-world relevance and implications of our findings.

Given that the practical applications represent a key contribution to our study and provide actionable insights for practitioners and the target population, we consider that keeping this section distinct allows readers to better appreciate its significance. The Limitations section, meanwhile, remains essential to critically reflect on the study’s constraints.

Therefore, we respectfully prefer to keep these sections separate, but we have revised their content to ensure clarity and coherence throughout the manuscript.

Round 2

Reviewer 1 Report

Comments and Suggestions for Authors

My opinion of this work has not changed following the review process. Although the work has improved in terms of formal issues, I still believe that a study based on the use of self-reported scales and with such a low number of participants does not provide sufficient scientific evidence to justify its publication in a Q1 journal.

Reviewer 2 Report

Comments and Suggestions for Authors

Thnak you for taking into m¡considretaion my suggestions.